# Discovery of Novel Delta Opioid Receptor (DOR) Inverse Agonist and Irreversible (Non-Competitive) Antagonists

**DOI:** 10.3390/molecules26216693

**Published:** 2021-11-05

**Authors:** Parthasaradhireddy Tanguturi, Vibha Pathak, Sixue Zhang, Omar Moukha-Chafiq, Corinne E. Augelli-Szafran, John M. Streicher

**Affiliations:** 1Department of Pharmacology, College of Medicine, University of Arizona, Tucson, AZ 85724, USA; parthasaradhit@arizona.edu; 2Department of Chemistry, Division of Drug Discovery, Southern Research Institute, Birmingham, AL 35205, USA; vibhabpathak@gmail.com (V.P.); szhang@southernresearch.org (S.Z.); omoukha-chafiq@southernresearch.org (O.M.-C.); caugelli-szafran@southernresearch.org (C.E.A.-S.)

**Keywords:** delta opioid receptor, inverse agonist, irreversible antagonist, non-competitive antagonist, molecular pharmacology

## Abstract

The delta opioid receptor (DOR) is a crucial receptor system that regulates pain, mood, anxiety, and similar mental states. DOR agonists, such as SNC80, and DOR-neutral antagonists, such as naltrindole, were developed to investigate the DOR in vivo and as potential therapeutics for pain and depression. However, few inverse agonists and non-competitive/irreversible antagonists have been developed, and none are widely available. This leaves a gap in our pharmacological toolbox and limits our ability to investigate the biology of this receptor. Thus, we designed and synthesized the novel compounds SRI-9342 as an irreversible antagonist and SRI-45128 as an inverse agonist. These compounds were then evaluated in vitro for their binding affinity by radioligand binding, their functional activity by ^35^S-GTPγS coupling, and their cAMP accumulation in cells expressing the human DOR. Both compounds demonstrated high binding affinity and selectivity at the DOR, and both displayed their hypothesized molecular pharmacology of irreversible antagonism (SRI-9342) or inverse agonism (SRI-45128). Together, these results demonstrate that we have successfully designed new inverse agonists and irreversible antagonists of the DOR based on a novel chemical scaffold. These new compounds will provide new tools to investigate the biology of the DOR or even new potential therapeutics.

## 1. Introduction

G-protein coupled receptors (GPCRs) are a superfamily of integral plasma membrane proteins and are involved in a broad array of signaling pathways and subsequent physiological processes. GPCRs are important drug targets, and over 25% of all approved drugs currently on the market are known to evoke their pharmaceutical effects through GPCRs [1]. The delta opioid receptor (DOR) is one such GPCR that has been linked to the regulation of pain, mood, depression, and similarly important brain states [2]. Numerous classical full/partial agonists, such as SNC80 and DPDPE, and neutral antagonists, such as naltrindole, have been developed or described for the DOR. However, there are far fewer inverse agonists and irreversible/non-competitive antagonists for this target.

A non-competitive antagonist is an insurmountable antagonist that can act either in one of two ways: via binding to an allosteric site of the receptor [3] or by irreversibly binding to the active site of the receptor. Although the mechanism of antagonism is different for both, they are called “non-competitive” since the end result of each are functionally the same. Unlike competitive antagonists, which commonly affect the amount of agonist necessary to achieve a maximal response but will not affect the magnitude of that maximal response, non-competitive antagonists decrease the level of the maximum response which can be accomplished by any amount of agonist. This unique property receives the name as “non-competitive” because the effects cannot be overcome no matter how much agonist is present. One example is the commonly used beta-funaltrexamine, which is selective for the mu opioid receptor [4]. Only a few such ligands have been reported for the DOR, and none are widely available. These include a naphthalene-dialdehyde modification of 6′-aminonaltrindole [5] and 5′-naltrindole-isothiocyanate [6].

By contrast, an inverse agonist does have intrinsic activity at the orthosteric site. However, unlike an agonist, an inverse agonist shifts the energy landscape to further disfavor the receptor active state and favor the inactive state. This suppresses baseline receptor activity to the extent that it is below that of the unliganded state [3]. The first such described ligand for the DOR was ICI-174864, a peptidic inverse agonist reported by Costa and Herz [7]. Later, additional DOR inverse agonists were reported, such as (+)−KF4 [8], naltrindole (NTI) derivatives [9], amide/sulfonamide substituted NTI [10], as well as other peptidic [11,12,13] and nonpeptidic [14,15,16,17] molecules.

Here, we report the discovery and characterization of a new DOR non-competitive antagonist (SRI-9342) and a new inverse agonist (SRI-45128). These ligands have strong selectivity for the DOR and potent functional activity in vitro. The discovery of these ligands further expands the limited pharmacological tools available to probe the DOR and could even provide potential future therapeutics.

## 2. Results

### 2.1. Rational Design of DOR Ligands

The DOR ligands SRI-9342 (irreversible antagonist) and SRI-45128 (inverse agonist) were designed based on our previously published computational docking studies of the SRI-9409 scaffold core [18], as well as DOR ligands reported in the study, including SYK-623 [19]. When this core binds to the orthosteric site of DOR, the right-hand side indole moiety would face the extracellular opening of the binding pocket and be adjacent to the functionally important K214 of DOR [20], which is also a potentially reactive residue. Thus, the idea of adding a warhead to the scaffold to further increase its binding affinity to DOR was explored, which resulted in SRI-9342. The α,β-unsaturated pyridin group of SRI-9342 is supposed to be in a proper position to form a covalent bond with the nitrogen on the side chain of K214 via Michael’s addition, turning it into an irreversible DOR antagonist. On the other hand, the cyclopropyl group of the scaffold would face the bottom of the binding pocket and form hydrophobic contact with the W274 of DOR, which is an important residue responsible for the switch between agonism and antagonism [20,21]. Therefore, chemical modifications were also explored at this position, which resulted in SRI-45128 with DOR inverse agonism. Notably, SRI-45128 was distinct from parent scaffold SRI-9409 in the sense that the introduction of carbonyl next to the cyclopropyl group (inspired by SYK-623) would reduce the basicity of the neighboring nitrogen of SRI-45128, which was no longer able to form a salt-bridge with key residue D128. In addition, SRI-45128 is distinguishable from SYK-623 in the sense that the former possessed a pyridine-4-phenylchloride moiety while the latter possessed an indole moiety on the right-hand side.

### 2.2. Synthesis of Novel DOR Irreversible Antagonist and Inverse Agonists

SRI-9342 (irreversible antagonist, Figure 1) was synthesized with a 44% yield via the reaction of naltrexone hydrochloride and *trans*-4-hydrazino-2-stilbazole dihydrochloride following the same procedures previously reported [22,23]. SRI-45128 (inverse agonist, Figure 2) was synthesized in nine steps by using the procedures reported in [24], subject to a few modifications. In our alternative route, the protecting group on the phenolic hydroxyl group of naltrexone (**1**) was changed from methyl to benzyl to achieve an overall improvement in yields. The 6-ketone group of compounds (**2**) was protected as 1,3-dioxolane, which was followed by acetylation of the 14-OH under refluxing in Ac_2_O. The cyclopropyl methyl group of the resulting acetate compound (**3**) was exchanged for a trichloroethoxycarbonyl group at 140 °C with an excess amount of trichloroethoxycarbonyl chloride to afford carbamate (**4**). The carbamate and the acetate group in compound (**4**) were further hydrolyzed with aqueous KOH at 110 °C to afford compound (**5**). The reaction of compound (**5**) with cyclopropyl carbonyl chloride in the presence of Et_3_N afforded compound (**6**) with an 89% yield. Removal of the 1,3-dioxolane group in amide (**6**) was done by using HCl and MeOH at reflux conditions to afford ketone (**7**). Finally, further deprotection of the benzyl group in (**7**), followed by annulation with 2-(4-chlorophenyl)-3-hydroxy-prop-2-enal in the presence of ammonium acetate, afforded SRI-45128 in two steps. We also synthesized SYK-623 for use as a control group (inverse agonist, Figure 3). This was achieved in two steps, from intermediate (**7**), by using the procedure reported in [19] with the following modifications: Compound (**7**) was reacted with phenyl hydrazine in acetic acid under reflux conditions followed by deprotection of the benzyl group to afford SYK-623. All compounds were confirmed for identity and high purity, which is sufficient for pharmacological characterization (see Methods).

### 2.3. All Compounds Display High DOR Binding Affinity

All synthesized compounds were evaluated for binding to the human DOR using competition radioligand binding. All compounds showed one site full competition, suggesting full occupancy of the orthosteric binding site (Figure 1). Notably, the compounds are also bound to the DOR with high affinity with K_I_ values of 4.9–24 nM (Figure 1). We also tested SRI-45128 and SYK-623 for binding to the MOR and KOR, which would provide insight into compound selectivity. The compounds bound very weakly to the MOR, showing incomplete curves even at 10 μM. This suggests that both compounds are at least 120-fold selective for DOR over MOR. The compounds bound slightly better to the KOR, providing near-complete curves and K_I_ values, ranging from 2000 to 2500 nM, suggesting that both compounds are at least 104-fold selective for DOR over KOR (Figure 1). These results demonstrate that these compounds are high-affinity DOR ligands, and both inverse agonists display strong DOR selectivity. The expected performance of SYK-623 further confirms our findings.

### 2.4. SRI-9342 Displays Irreversible Antagonism at the DOR at Low Concentrations

Now that we showed that these compounds bound with high affinity to the DOR, we next sought to evaluate their functional activity. We tested the putative irreversible antagonist SRI-9342 for this activity using ^35^S-GTPγS coupling. We ran SNC80 concentration curves with fixed and increasing concentrations of SRI-9342 present in each successive SNC80 curve. At 0.1 nM and 1 nM SRI-9342, we found that the SNC80 potency was actually better than the potency of SNC80 alone, while the efficacy was successively reduced to 92% and 84% (Figure 2). This behavior fits with the expected behavior of an irreversible antagonist, where potency is maintained at least initially, while efficacy is reduced. At higher concentrations (10–1000 nM), we observed large and successive shifts in both potency and measured efficacy, which is consistent with increasing receptor loss from the system. At 10,000 nM, we observed a puzzling partial recovery of both potency and efficacy. This may represent a non-specific effect at high concentrations (Figure 2). Overall, these findings are consistent with SRI-9342 displaying irreversible antagonism at low concentrations.

### 2.5. SRI-45128 Displays DOR Inverse Agonism

Similar to SRI-9342, we sought to evaluate the inverse agonist functional activity of SRI-45128 with SYK-623 as a comparison control. The GTPγS assay used above has a generally low baseline receptor activity level, at least for the opioid receptors, so we switched assays to a live cell cAMP accumulation assay. This assay uses forskolin to stimulate cAMP levels, which are then inhibited/suppressed by the Gα_I_-coupled activity of the DOR. As expected, SNC80 demonstrated potent and efficacious suppression of cAMP levels in DOR-CHO cells, which is in line with the expected activity of the receptor (Figure 3). By contrast, both SRI-45128 and SYK-623 showed efficacious inverse agonist activity and actually boosted cAMP levels, which is consistent with suppressing the baseline activity of the DOR (Figure 3). This activity was also efficacious, with an E_MAX_ of −67% and −56%, respectively. These results suggest that SRI-45128 is a robust inverse agonist, further confirmed by the performance of the SYK-623 comparison control.

## 3. Discussion

As noted in the Introduction, a limited set of DOR irreversible antagonists and inverse agonists has been discovered and reported [5,11,12,13,14,15,16,17,19,25,26]. The ligands we report here were developed from a novel naltrexone scaffold and thus represent a significant contribution to the limited set of pharmacological tools available to probe the DOR. These ligands will help build the structure–activity relationship of irreversible antagonism and inverse agonism, and they could be used to investigate DOR function in vivo. 

These novel compounds also have some possibility to inform future therapeutic candidates to target the DOR. For example, DOR activation has been associated with Alzheimer’s disease, and DOR antagonism was shown to prevent and reverse Alzheimer’s pathology in a mouse model [27]. Considering the long time-scales for Alzheimer’s treatment, especially in a prevention paradigm, a long-lasting irreversible antagonist could be of considerable therapeutic benefit versus a short-acting competitive antagonist. Alternatively, an inverse agonist could be more effective than a standard neutral antagonist. DOR antagonists/inverse agonists have also not been associated with seizure activity as for some DOR agonists, suggesting their improved safety vs. agonists. This suggests that the active development of functionally selective DOR agonists to avoid seizures should not be necessary for these compounds [28,29]. However, one caution is that these compounds have only been tested for brief exposures in vitro and could thus possess other toxic effects. This will have to be examined in future studies.

Future works should also investigate these compounds in greater detail. Based on our binding studies, it is clear that all compounds bind selectively and with a high affinity to the DOR orthosteric site. However, the functional studies were not quite as clear. SRI-9342 displayed clear signs of irreversible antagonism at 0.1 and 1 nM. However, 10 nM caused a rapid loss of potency that continued at 100 nM and 1000 nM. The compound could be a full irreversible antagonist, and the activity at 10 nM could represent a rapid loss of receptors from the system that would eventually lead to the same reduction in potency as with other irreversible antagonists. Alternatively, the compound could have mixed activity, with irreversible antagonism at low concentrations and different functional activity, similar to competitive antagonism, at high concentrations. This mixed activity has been observed with other compounds, such as naloxonazine [30]. SRI-45128 also displayed clear inverse agonist activity. However, the potency of this activity was considerably less than the binding affinity. This could represent the poor intrinsic efficacy of the compounds, or it could represent a relatively insensitive system for baseline receptor suppression. These details should be investigated for these compounds, working out the exact mechanisms of action and activity in different DOR-related signaling systems (e.g., ERK-MAPK activation instead of cAMP signaling). In addition, all compounds should be investigated for in vivo activity and whether the in vivo testing matches the predictions made via the in vitro testing reported here.

## 4. Materials and Methods

### 4.1. Chemical Synthesis and Characterization

All solvents and reagents were used as purchased without further purification. Unless otherwise stated, reactions were carried out under nitrogen atmosphere. Reaction conditions and yields were not optimized. The progress of all reactions was monitored by thin-layer chromatography (TLC) on pre-coated silica gel (60F254) aluminum plates (0.25 mm) from E. Merck and visualized using UV light (254 nm). Purification of compounds was performed on an Isco Teledyne Combiflash Rf200 with four channels to carry out sequential purification. Universal RediSep solid sample loading pre-packed cartridges (5.0 g silica) were used to absorb the crude product and purified on 12 g silica RediSep Rf Gold Silica (20–40 μm spherical silica) columns using appropriate solvent gradients. Melting points were determined in open capillary tubes with a Thomas–Hoover melting point apparatus or SRS OptiMelt automated melting point system and are uncorrected. High-resolution mass spectrometry (HRMS) or liquid chromatography with tandem mass spectrometry (LC-MS/MS) analysis was performed with an Agilent 1100 LC-MS TOF instrument using electrospray ionization (ESI) or with Agilent 1290 ultra-performance liquid chromatography (UPLC)/Sciex Triple Quad 6500+. ^1^H NMR spectra were recorded at 400 MHz on an Agilent/Varian MR-400 spectrometer, and ^13^C NMR spectra were recorded either at 100.574 MHz on an Agilent/Varian MR-400 spectrometer or at 125.76 MHz on a Bruker Avance III-HD 600 MHz Spectrometer. The chemical shifts (δ) are reported in parts per million (ppm) and referenced according to the deuterated solvent for ^1^H spectra (CDCl_3_, 7.26, DMSO-*d_6_*, 2.50, or TMS 0.0) and ^13^C spectra (CDCl_3_, 77.2 or DMSO-*d_6_*, 39.5). The purity of the final compounds was checked by analytical HPLC using an Agilent 1100 LC system equipped with a phenomenex Kinetex C18 column (5 μm, 4.6 × 150 mm) and a diode array detector (DAD) using the solvent system: solvent A: H_2_O/0.1% trifluoroacetic acid, solvent B: CH_3_CN/0.1% trifluoroacetic acid, 0–95% B over 22 min, flow rate 1 mL/min, λ 254 nm and λ 280 nm (System 1) or using a Waters HPLC system equipped with a Sunfire C18 column (5 μm, 4.6 × 150 mm) and a Waters 2998 photodiode array detector using the solvent system: solvent A: H_2_O/0.1% formic acid, solvent B: CH_3_CN/0.1% formic acid, 10–90% B over 20 min, flow rate 2 mL/min, λ 254 nm (System 2) or using an Agilent 1200 LC system equipped with phenomenex Kinetex Phenyl-Hexyl column (2.6 μm, 4.6 × 50 mm) and a diode array detector (DAD) using the solvent system: solvent A: H_2_O/0.1% formic acid, solvent B: CH_3_CN/0.1% formic acid, 0–95% B over 4.5 min, flow rate 2 mL/min, λ 254 nm (System 3). On the basis of NMR, HPLC-DAD, and HRMS (mass error less than 5 ppm), all final compounds were ≥95% pure.

#### 4.1.1. (4b*S*,8*R*,8a*S*,14b*R*)-7-(Cyclopropylmethyl)-11-((E)-2-(pyridin-2-yl)vinyl)-5,6,7,8,14,14b-hexahydro-4,8-methanobenzofuro [2,3-a]pyrido[4,3-b]carbazole-1,8a(9*H*)-diol (SRI-9342)

This compound was synthesized as previously described [22,23].

#### 4.1.2. ((4b*S*,8*R*,8a*S*,13b*R*)-11-(4-Chlorophenyl)-1,8a-dihydroxy-5,6,8,8a,9,13b-hexahydro-7H-4,8-methanobenzofuro[3,2-h]pyrido[3,4-g]quinolin-7-yl)(cyclopropyl)methanone (SRI-45128)

This compound was synthesized by a modified procedure of the reported method [24] as described below.

#### 4.1.3. 2,2,2-Trichloroethyl (4′*R*,7a′*R*,12b′*S*)-9′-(benzyloxy)-1′,2′,4′,6′-tetrahydro-3′*H*,7a′*H*-spiro[[1,3]dioxolane-2,7′-[4,12]methanobenzofuro[3,2-e]isoquinoline]-3′-carboxylate (**2**)

To a suspension of naltrexone (**1**) (5 g, 14.6 mmol) in acetone (200 mL), benzyl bromide (2.6 mL, 21.9 mmol) and potassium carbonate (4.0 mg, 29.3 mmol) were added. The reaction mixture was refluxed for 2 h; then, it was filtered, and the solid was washed with acetone (50 mL). The filtrate was concentrated under reduced pressure to afford a crude white solid, which was purified by the column chromatography over a column of silica gel, using EtOAc:Hexane, 1:3 as an eluant, to afford compound (**2**) (5.8 g, 58%) as a white solid. ^1^H NMR (400 MHz, CDCl_3_) δ 7.47–7.43 (m, 2H), 7.34 (ddt, *J* = 8.1, 6.5, 1.7 Hz, 2H), 7.31–7.26 (m, 1H), 6.71 (d, *J* = 8.2 Hz, 1H), 6.58–6.54 (m, 1H), 5.29 (d, *J* = 12.0 Hz, 1H), 5.21 (d, *J* = 8.9 Hz, 1H), 4.69 (s, 1H), 3.17 (d, *J* = 5.9 Hz, 1H), 3.09–2.98 (m, 2H), 2.69 (ddt, *J =* 12.1, 5.3, 1.3 Hz, 1H), 2.56 (ddd, *J* = 18.5, 6.1, 1.1 Hz, 1H), 2.47–2.38 (m, 3H), 2.31 (dt, *J* = 14.4, 3.2 Hz, 1H), 2.13 (td, *J* = 12.2, 3.8 Hz, 1H), 1.93–1.86 (m, 1H), 1.71–1.52 (m, 3H), 0.92–0.83 (m, 1H), 0.59–0.51 (m, 2H), 0.16–0.11 (m, 2H). ESI MS *m*/*z* 432 [M + H]^+^.

#### 4.1.4. (4′*R*,4a′*S*,7a′*R*,12b′*S*)-9′-(Benzyloxy)-3′-(cyclopropylmethyl)-1′,2′,3′,4′,5′,6′-hexahydro-4a′*H*,7a′*H*-spiro[[1,3]dioxolane-2,7′-[4,12]methanobenzofuro[3,2-e]isoquinolin]-4a′-yl acetate (**3**)

To a solution of compound (**2**) (5.0 g, 11.6 mmol) in toluene (50 mL) were added *p*-TsOH.H_2_O (3.0 g, 17.4 mmol) and ethylene glycol (3.9 mL, 69.5 mmol), and the mixture was refluxed with a Dean–Stark apparatus for 17 h under an argon atmosphere. After cooling to room temperature, the reaction mixture was basified with potassium carbonate (3 g) and saturated aqueous NaHCO_3_ solution (30 mL); then, it was extracted with CHCl_3_ (3 × 100 mL). The organic layer was washed with brine, dried over Na_2_SO_4_, and concentrated under reduced pressure to afford a crude product as a colorless solid. The crude product was suspended in Ac_2_O (35 mL), and the mixture was refluxed for 1 h under an argon atmosphere. After cooling to room temperature, the reaction mixture was concentrated and co-evaporated with toluene three times. The obtained residue was purified by column chromatography on silica gel using 0–5% MeOH in DCM to afford the desired compound (**3**) (3.8 g, 87% in two steps) as a white solid. ^1^H NMR (400 MHz, CDCl_3_) δ 7.42–7.27 (m, 5H), 6.86 (d, *J* = 8.2 Hz, 1H), 6.66 (d, *J* = 8.3 Hz, 1H), 5.36 (d, *J* = 5.2 Hz, 1H), 5.21–5.10 (m, 2H), 4.67 (s, 1H), 4.17 (ddd, *J* = 7.4, 6.6, 5.3 Hz, 1H), 4.01 (q, *J* = 6.6 Hz, 1H), 3.89 (dt, *J* = 7.4, 6.4 Hz, 1H), 3.78 (td, *J* = 6.6, 5.3 Hz, 1H), 3.46–3.32 (m, 2H), 3.23–3.06 (m, 2H), 3.02–2.89 (m, 2H), 2.66 (dt, *J* = 13.8, 3.2 Hz, 1H), 2.50 (s, 4H), 1.85 (td, *J* = 14.4, 2.8 Hz, 1H), 1.57 (dddd, *J* = 11.4, 8.5, 6.0, 3.2 Hz, 3H), 1.13–1.02 (m, 2H), 0.82–0.63 (m, 2H), 0.45 (ddt, *J* = 8.7, 5.7, 4.5 Hz, 1H). ESI MS *m*/*z* 518 [M + H]^+^.

#### 4.1.5. 2,2,2-Trichloroethyl (4′*R*,4a′*S*,7a′*R*,12b′*S*)-4a′-acetoxy-9′-(benzyloxy)-1′,2′,4′,4a′,5′,6′-hexahydro-3′*H*,7a′*H*-spiro[[1,3]dioxolane-2,7′-[4,12]methanobenzofuro[3,2-e]isoquinoline]-3′-carboxylate (**4**)

To a solution of compound (**3**) (3.0 g, 5.8 mmol) in 1,1,2,2-tetrachloroethane (25 mL) were added K_2_CO_3_ (5 g, 36.5 mmol) and 2,2,2-trichloroethyl chloroformate (5 mL, 36.5 mmol), and the mixture was stirred at 140 °C for 14 h under argon atmosphere. The reaction mixture was cooled to room temperature, and H_2_O (20 mL) was added. The mixture was extracted with CHCl_3_ (2 × 100 mL). The organic layer was washed with brine, dried over Na_2_SO_4_, and concentrated under reduced pressure to give a crude residue, which was purified by column chromatography on silica gel using 0–10% hexanes:EtOAc to afford compound (**4**) (2.1 g, 56%) as a light yellow amorphous solid. ^1^H NMR (400 MHz, CDCl_3_) δ 7.42 (tdd, *J* = 5.0, 4.0, 1.8 Hz, 2H), 7.39–7.28 (m, 3H), 6.83 (dd, *J* = 8.3, 1.3 Hz, 1H), 6.61 (dd, *J* = 8.3, 4.8 Hz, 1H), 5.62 (t, *J* = 4.5 Hz, 1H), 5.16 (q, *J* = 12.0 Hz, 2H), 4.93–4.85 (m, 1H), 4.70–4.63 (m, 1H), 4.60 (s, 1H), 4.21–4.15 (m, 1H), 4.07–3.98 (m, 2H), 3.91–3.86 (m, 1H), 3.80 (td, *J* = 6.6, 5.2 Hz, 1H), 3.09 (dt, *J* = 18.2, 6.0 Hz, 1H), 2.98–2.73 (m, 3H), 2.39 (tt, *J* = 12.6, 5.3 Hz, 1H), 2.07 (s, 1H), 2.05 (s, 1H), 1.86–1.77 (m, 1H), and 1.55–1.48 (m, 2H). ESI MS *m*/*z* 579.8 [M-AcOH]^+^.

#### 4.1.6. (4′*R*,4a′*S*,7a′*R*,12b′*S*)-9′-(Benzyloxy)-1′,2′,3′,4′,5′,6′-hexahydro-4a′*H*,7a′*H*-spiro[[1,3]dioxolane-2,7′-[4,12]methanobenzofuro[3,2-e]isoquinolin]-4a′-ol (**5**)

To a suspension of compound (**4**) (1.5 g, 2.6 mmol) in DMSO (20 mL) was added 12*M* aqueous KOH solution (7 mL), and the mixture was stirred for 6 h at 110 °C under an argon atmosphere. After cooling to room temperature, the reaction mixture was adjusted to pH 10 with saturated aqueous NH_4_Cl solution (20 mL) and extracted with a mixed solution, *i*-PrOH/CHCl_3_ = 1:3 (3 × 10 mL). The organic layer was washed with brine, dried over Na_2_SO_4_, and concentrated under reduced pressure to afford a crude residue, which was purified by chromatography on silica gel using 10% MeOH in DCM + 0.1% NH_3_ to generate compound (**5**) (0.7 g, 64%) as a colorless solid. ^1^H NMR (400 MHz, DMSO-*d*_6_) δ 7.42–7.33 (m, 4H), 7.31–7.27 (m, 1H), 6.75 (d, *J* = 8.2 Hz, 1H), 6.52 (dd, *J* = 8.2, 0.8 Hz, 1H), 5.12–5.02 (m, 2H), 4.84 (s, 1H), 4.33 (s, 1H), 4.02–3.97 (m, 1H), 3.83 (q, *J* = 6.4 Hz, 1H), 3.78–3.66 (m, 2H), 2.89–2.78 (m, 3H), 2.55–2.49 (m, 1H), 2.34 (td, *J* = 12.6, 3.8 Hz, 1H), 2.15–1.98 (m, 2H), 1.40–1.28 (m, 3H), and 1.08–1.02 (m, 1H).

#### 4.1.7. ((4′*R*,4a′*S*,7a′*R*,12b′*S*)-9′-(Benzyloxy)-4a′-hydroxy-1′,2′,4′,4a′,5′,6′-hexahydro-3′*H*,7a′*H*-spiro[[1,3]dioxolane-2,7′-[4,12]methanobenzofuro[3,2-e]isoquinolin]-3′-yl)(cyclopropyl)methanone (**6**)

To a stirred solution of compound (**5**) (500 mg, 1.14 mmol) in CH_2_Cl_2_ (12 mL) were added Et_3_N (0.24 mL, 1.71 mmol) and cyclopropanecarbonyl chloride (143 mg, 1.37 mmol) at 0 °C under an argon atmosphere. After stirring for 2 h at room temperature, the reaction mixture was diluted with CH_2_Cl_2_ (10 mL) and washed with saturated aqueous NaHCO_3_ solution (20 mL). The organic layer was washed with brine, dried over Na_2_SO_4_, and concentrated under reduced pressure to give a crude residue, which was purified by column chromatography on silica gel (80–100% EtOAc in n-hexane) to afford compound (**6**) (500 mg, 89%) as a colorless amorphous product. ^1^H NMR (400 MHz, CDCl_3_) δ 7.45–7.41 (m, 2H), 7.39–7.28 (m, 3H), 6.81 (d, *J* = 8.2 Hz, 1H), 6.59 (d, *J* = 8.0 Hz, 1H), 5.17 (q, *J* = 12.0 Hz, 2H), 4.94 (d, *J* = 5.9 Hz, 1H), 4.54 (d, *J* = 19.1 Hz, 1H), 4.40 (d, *J* = 21.2 Hz, 1H), 4.19–4.12 (m, 1H), 4.11–3.78 (m, 4H), 3.22–2.77 (m, 3H), 2.67–2.34 (m, 2H), 2.17 (t, *J* = 10.8 Hz, 1H), 1.71–1.45 (m, 5H), 1.10–0.92 (m, 2H), 0.80 (d, *J* = 9.8 Hz, 2H).

#### 4.1.8. (4*R*,4a*S*,7a*R*,12b*S*)-9-(Benzyloxy)-3-(cyclopropanecarbonyl)-4a-hydroxy-2,3,4,4a,5,6-hexahydro-1H-4,12-methanobenzofuro[3,2-e]isoquinolin-7(7a*H*)-one (**7**)

A mixture of compound (**6**) (0.10 g, 0.20 mmol) in 1 *M* HCl (0.8 mL) was stirred for 15 h at 80 °C under an argon atmosphere. After cooling to room temperature, the reaction mixture was poured in 4*N* NaOH solution at 0 °C and extracted with CHCl_3_, dried over Na_2_SO_4_, filtered, and concentrated to dryness to afford the crude material. The crude residue was purified by column chromatography on silica gel (5–15% (28% NH_3_ aq/MeOH = 1:9) in CHCl3) to afford intermediate (**7**) (75 mg, 82%) as a colorless amorphous product with some impurities which was used as such in next step. ESI MS *m*/*z* 446 [M + H]^+^.

#### 4.1.9. ((4b*S*,8*R*,8a*S*,13b*R*)-11-(4-Chlorophenyl)-1,8a-dihydroxy-5,6,8,8a,9,13b-hexahydro-7*H*-4,8-methanobenzofuro[3,2-h]pyrido[3,4-g]quinolin-7-yl)(cyclopropyl)methanone (SRI-45128)

Compound (**7**) (150 mg, 0.3400 mmol) was dissolved in methanol (8 mL), and 10% palladium black (35.8 mg, 0.03 mmol) was added to the reaction. The reaction mixture was stirred under H_2_ atmosphere (balloon) for 2 h at room temperature. The reaction mixture was filtered through a Celite pad, washed with MeOH (20 mL), and concentrated. To the solution of crude material in acetic acid (2 mL), ammonium acetate (43.4 mg, 0.56 mmol), and (*E*)-2-(4-chlorophenyl)-3-hydroxy-prop-2-enal (77.07 mg, 0.42 mmol) were added, and the reaction mixture was refluxed overnight. The reaction mixture was cooled to room temperature, and acetic acid was removed. The obtained residue was diluted with water, neutralized with ammonium hydroxide to pH 7, extracted with DCM (3 × 10 mL), dried over Na_2_SO_4_, and purified by column chromatography using 0–10% MeOH in DCM to give a solid material, which was further purified by a preparative plate using 9:1 CHCl3-MeOH to afford SRI-45128 (35 mg, 25%) as a white solid. TLC (10% MeOH/DCM): R*f =* 0.35; ^1^H NMR (400 MHz, CD_3_OD) δ 8.60 (dd, *J =* 5.1, 2.2 Hz, 1H), 7.58 (dd, *J* = 3.6, 2.3 Hz, 1H), 7.50–7.44 (m, 2H), 7.42–7.37 (m, 2H), 6.66–6.58 (m, 2H), 5.42 (d, *J* = 2.7 Hz, 1H), 5.05 (d, *J* = 6.5 Hz, 0.5H), 4.74 (d, *J* = 6.5 Hz, 0.5H), 4.51–4.42 (m, 0.5H), 4.23 (dd, *J* = 14.0, 5.1 Hz, 0.5H), 3.38–3.30 (m, 1H), 3.28–3.15 (m, 1H), 2.88–2.75 (m, 2H), 2.55 (td, *J* = 12.8, 5.2 Hz, 0.5H), 2.40 (td, *J* = 12.8, 5.3 Hz, 0.5H), 2.33–2.22 (m, 1H), 2.16–1.96 (m, 1H), 1.73–1.53 (m, 2H), 1.20–1.06 (m, 1H), 1.02–0.93 (m, 1H), 0.89–0.80 (m, 3H). ^13^C NMR (101 MHz, CD_3_OD) δ 173.95, 145.50, 139.99, 136.28, 136.22, 135.33, 135.19, 134.23, 131.31, 128.88, 128.23, 128.19, 123.58, 119.41, 119.34, 117.64, 88.88, 71.56, 71.38, 47.38, 47.16, 47.01, 46.95, 46.94, 38.56, 36.28, 35.04, 32.45, 31.83, 29.71, 29.04, 11.06, 10.99, 6.91, 6.59, 6.53, 6.42. HRMS (ESI) *m/z* calcd for C_29_H_25_ClN_2_O_4_ [M + H]^+^: 501.1576, found: 501.1570; HPLC (system 1) *t*_R_ = 10.7 min, purity = 99%.

#### 4.1.10. Cyclopropyl((4bS,8R,8aS,14bR)-1,8a-dihydroxy-5,6,8a,9,14,14b-hexahydro-4,8-methanobenzofuro[2,3−a]pyrido [4,3−b]carbazol-7(8H)-yl)methanone (SYK-623)

This compound was prepared by a modification of the reported protocol [19]. A solution of compound (**7**) (200 mg, 0.45 mmol) in acetic acid (5 mL) was supplemented with phenylhydrazine (58.3 mg, 0.54 mmol), and the reaction mixture was refluxed for 2 h. After cooling to room temperature, the reaction mixture was concentrated and co-evaporated with toluene to afford a crude product. Saturated Na_2_CO_3_ was added, and the mixture was extracted with methylene chloride, dried over sodium sulfate, concentrated, chromatographed on an Isco Combiflash system using 0–5% MeOH in DCM to give a white solid which was dissolved in methanol (10 mL), and then 10% palladium black (24.6 mg, 0.02 mmol) was added to the reaction. The mixture was stirred under H_2_ atmosphere (balloon) for 2 h at room temperature and filtered through celite. The celite pad was washed with MeOH (20 mL), and the filtrate was evaporated to give a white residue, which was purified by column chromatography using 0–10% MeOH in DCM to afford SYK-623 (50 mg, 48%) as a white solid. TLC (10% MeOH/DCM): R*_f_ =* 0.35; ^1^H NMR (400 MHz, CD_3_OD) δ 7.36 (ddt, *J =* 7.8, 6.7, 1.0 Hz, ^1^H), 7.30 (ddt, *J =* 8.2, 4.1, 0.9 Hz, ^1^H), 7.07 (dddd, *J* = 8.2, 7.0, 3.8, 1.2 Hz, ^1^H), 6.94 (dddd, *J* = 8.0, 7.0, 4.7, 1.0 Hz, ^1^H), 6.63–6.51 (m, ^2^H), 5.61 (s, ^0.5^H), 5.07 (d, *J* = 6.6 Hz, ^0.5^H), 4.75 (d, *J* = 6.6 Hz, ^0.5^H), 4.49–4.38 (m, ^0.5^H), 4.18–4.08 (m, ^0.5^H), 3.44 (dd, *J =* 18.5, 6.7 Hz, ^0.5^H), 3.33–3.25 (m, ^1^H), 3.15 (td, *J* = 13.4, 3.8 Hz, ^0.5^H), 3.00–2.58 (m, ^4^H), 2.51 (td, *J* = 12.8, 5.2 Hz, ^0.5^H), 2.07 (tt, *J* = 7.9, 4.8 Hz, ^0.5^H), 1.96–1.88 (m, ^0.5^H), 1.71–1.62 (m, ^1^H), 1.00–0.90 (m, ^1^H), 0.89–0.74 (m, ^3^H). HRMS (ESI) *m*/*z* calcd for C_26_H_24_N_2_O_4_ [M + H]^+^: 429.1810, found: 429.1801; HPLC (system 1) *t*_R_ = 10.7 min, purity = 96%.

### 4.2. Cell Culture

A Chinese Hamster Ovary (CHO-K1) parental cell line expressing the human DOR was obtained from PerkinElmer (#RBHODM-K) and used for all experiments. Cells were maintained in 1:1 DMEM/F12 media with 1× penicillin/streptomycin supplement and 10% heat-inactivated fetal bovine serum (all from Invitrogen/ThermoFisher) in a 37 °C/5% CO_2_ incubator. Maintenance cultures were further supplemented with 500 μg/mL of G418 to preserve receptor selection/expression (Invitrogen/ThermoFisher). Cells were generally passaged at 1:10 every 2 days. Similar cell lines for the human mu opioid receptor (PerkinElmer, #ES-542-C) and human kappa opioid receptor (PerkinElmer, #ES-541-C, Cambridge, MA, USA) were used for selectivity experiments and maintained as above.

### 4.3. Competition Radioligand Binding

Radioligand binding was generally carried out as reported in our previous work (e.g., [18,31,32,33]). Briefly, 30 μg of cell membrane protein was combined with 0.89 nM of ^3^H-diprenorphine (PerkinElmer, #NET1121250UC) and concentration curves of experimental ligand or reference drug (see Figure Legends for details) in a 200 μL reaction volume. The reactions were incubated at room temperature for 1 h; then, they were collected onto GF/B filter plates (PerkinElmer) using a Brandel Cell Harvester. Then, the plates were read on a PerkinElmer Microbeta2 scintillation counter. The resulting data were normalized to vehicle-alone treatment (100%) and non-specific binding with 10 μM reference compound (0%) and fit to a one-site competition binding model using GraphPad Prism 9.0. The previously measured K_D_ of the diprenorphine in each cell line [34] was used to calculate the K_I_ of each ligand at each receptor.

### 4.4. ^35^S-GTPγS Coupling

The ^35^S-GTPγS coupling assay was also performed as in our previous work (e.g., [18,31,35]). Briefly, 15 μg of cell membrane protein was combined with 0.1 nM of ^35^S-GTPγS (PerkinElmer, #NEG030H250UC) and concentration curves of ligand and SNC80 (see the figure legends for details) in a 200 μL reaction volume in the presence of 40 μM GDP. The reactions were incubated at 30 °C for 1 h; then, they were collected and read as above. The resulting data were normalized to stimulation caused by vehicle (0%) and 10 μM SNC80 (100%) and fit to a 3-variable (Hill Slope = 1) agonist model using GraphPad Prism 9.0.

### 4.5. cAMP Accumulation Assay

This assay was also carried out as reported in our previous work [31]. First, 20,000 cells/well were plated in a 96-well plate in growth medium as above for 24 h. Then, cells were serum-starved in DMEM/F12 for 4 h and then incubated with 500 μM IBMX for 20 min. Stimulation buffer contained 500 μM IBMX and 50 μM forskolin, which is a known cAMP inducer. Serial dilutions of SNC80, a known reference DOR agonist, or test compounds were added in stimulation buffer for 10 min. Then, incubation mixtures were halted by adding ice-cold assay buffer and heating the plate at 80 °C for 10 min. The plate was centrifuged at 4,000 rpm for 10 min at 4 °C; then, supernatants were transferred into a new 96-well plate. The supernatants were co-incubated with 1 pmol of ^3^H-cAMP (PerkinElmer #NET1161250UC) and 7 μg of bovine protein kinase A (Sigma-Aldrich, St. Louis, MO, USA) in 0.05% bovine serum albumin for 1 h at room temperature. Then, the reactions were collected and measured as above. The data were normalized to cAMP suppression caused by vehicle (0%) or 10 μM SNC80 (100%) and fit to a 3-variable (Hill Slope = 1) agonist curve by GraphPad Prism 9.0.

### 4.6. Data Analysis

The data generated by the above pharmacological analyses include binding affinity (K_I_) and functional potency/efficacy (EC_50_/E_MAX_). Each experiment was performed as 3 independent experiments using separate plates, drug dilutions, etc. (N = 3). The above values were calculated separately for each independent experiment and then reported as the mean ± SEM of the N = 3 set. Statistical comparisons are not typical for this sort of pharmacological characterization and were not employed here.

## Data Availability

All data sets are available from the Corresponding Author upon request.

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
