# Peer review of "Discovery of Novel Delta Opioid Receptor (DOR) Inverse Agonist and Irreversible (Non-Competitive) Antagonists"

_molecules, 2021, doi:10.3390/molecules26216693_

Round 1

Reviewer 1 Report

The current paper describes the design, synthesis of new DOR agonist/antagonist and their in vitro evaluation.

The introduction is well written and informative. 

In the section 2.1 the Authors could provide more detailed information concerning the synthesis and identification of intermediates and final products.

Presentation of Scheme 1 should be improved - in the second line instead of writing the only the number 5 the whole structure should be presented. Or an arrow of stage 7 should follow directly structure 5 in the first line.

Line 87-91 - what does 91%, 87%, 57%.... and so on in the footnote of scheme 1 mean? Probably it is yield, but it should be clarified. This remark concerns the whole footnote. 

Some grammar and spelling mistakes should be corrected.

The Authors did not provide any supplementary data, and therefore the manuscript lack NMR, MS spectra of synthesized compounds. It is not mandatory, but would improve the overall quality and merit of  the manuscript. 

Reviewer 2 Report

Review Report

Manuscript ID: molecules-1422678

The study titled “Discovery of Novel Delta Opioid Receptor (DOR) Inverse Agonist and Irreversible (Non-Competitive) Antagonists”. The design of work and the results of the investigation Delta Opioid Receptor (DOR) ligands seems impressive and has major impact in the field of drug discovery. It appears the chemistry and pharmacology data are of high quality and most of the major conclusions made by the authors are decently sounding. However, this paper will have a excellent impact on the field of drug discovery and is of high quality. It is indeed an important goal to discover high affinity and functionally selective Delta Opioid Receptor ligands. Nevertheless, there are minor errors with this paper that preclude its publication at this time.

“The authors stated that, the Delta Opioid Receptor (DOR) is a crucial receptor system that regulates pain, mood, anxiety, and similar mental states. DOR agonists, such as SNC80, and DOR neutral antagonists, such as naltrindole, have been developed to investigate the DOR in vivo, and as potential therapeutics for pain and depression. However, few inverse agonists and non-competitive/irreversible antagonists have been developed, and none are widely available. This leaves a gap in our pharmacological toolbox and limits our ability to investigate the biology of this receptor. Also, designed and synthesized the novel compounds SRI-9342 as an irreversible antagonist and SRI-45127/SRI-45128 as inverse agonists. These compounds were then evaluated in vitro for their binding affinity by radioligand binding and functional activity by 35S-GTPγS coupling and cAMP accumulation in cells expressing the human DOR. These new compounds will provide new tools to investigate the biology of the OR or even new potential therapeutics.”

The manuscript is well written, however, there are some (mostly minor) questions and suggestions for the chemistry part:

Comments and Concerns:

  1. Based on the pharmacology results, these compounds look promising. However, the authors must address the rational design of this study. Also suggested to add functional selectivity impact on the safer medications to treat neurological disorders.
  2. Scheme 1 step h), product yield missed should be similar to scheme 2 step h). (48%?)
  3. In the chemistry experimental part all the compounds absolute configurations RS should be Italicized. Also add space between the serial number and chemical name of the compound.
  4. Typo errors can be resolved in the experimental section.

Reviewer 3 Report

In this work, the authors designed and synthesized three antagonists of Delta opioid receptor. The binding affinity and functional activity were also assessed in vitro. I believe these compounds can expand the toolbox for modulation of DOR activity. As such, publication is recommended after following concerns are addressed.

  1. Before the illustration of the synthetic route to the targeted compounds, I suggest the author to incorporate a brief introduction to discuss the rational design of these structures.
  2. For the irreversible binding of SRI-9342, I think the Michael acceptor in the structure plays an important role. Is there any evidence to show the irreversible linkage between the 9342 and DOR? How toxic this compound is toward normal cell lines?
  3. For some compounds, such as 4, the MS data is missing. The resolution form of the MS data is not consistent.
